



# Sensitivity of modeled snow grain size retrievals to solar geometry, snow particle asphericity, and snowpack impurities

Zachary Fair[1], Mark Flanner[1], Adam Schneider[2], and S. McKenzie Skiles[3]

[1]Department of Climate and Space Sciences and Engineering, University of Michigan, Ann Arbor, MI
[2]Department of Earth System Science, University of California, Irvine, Irvine, CA
[3]Department of Geography, University of Utah, Salt Lake City, UT

**Correspondence:** Zachary Fair (zhfair@umich.edu)

**Abstract.** Snow grain size is an important metric to determine snow age and metamorphism, but it is difficult to measure. The effective grain size can be derived from spaceborne and airborne radiance measurements due to strong attenuation of near-infrared energy by ice. Consequently, a snow grain size inversion technique that uses hyperspectral radiances and exploits variations in the 1.03 $\mu$m ice absorption feature was previously developed for use with airborne imaging spectroscopy. Previ-

ous studies have since demonstrated the effectiveness of the technique, though there has yet to be a quantitative assessment of the retrieval sensitivity to snowpack impurities, ice particle shape, or solar geometry. In this study, we use the Snow, Ice, and Aerosol Radiative (SNICAR) model and a Monte Carlo photon tracking model to examine the sensitivity of snow grain size retrievals to changes in dust and black carbon content, anisotropic reflectance, changes in solar illumination angle ($\theta_0$), and scattering asymmetry parameter ($g$) associated with different particle shapes. Our results show that changes in these variables

can produce large grain size errors, especially when the effective grain size exceeds 500 $\mu$m. Dust content of 1000 ppm induces errors exceeding 800 $\mu$m, with the highest biases associated with small particles. Aspherical ice particles and perturbed solar zenith angles produce maximum biases of ∼540 $\mu$m and ∼400 $\mu$m respectively, when spherical snow grains and $\theta_0 = 60°$ are assumed in the generation of the retrieval calibration curve. Retrievals become highly sensitive to viewing angle when reflectance is anisotropic, with biases exceeding 1000 $\mu$m in extreme cases. Overall, we show that a more detailed understanding

of snowpack state and solar geometry improves the precision when determining snow grain size through hyperspectral remote sensing.

## 1 Introduction

The optical grain size of snow ($r_{eff}$) is a critical factor in the determination of snowpack albedo and metamorphism. The term

"optical grain size" does not refer to the actual size of individual snowflakes, but instead represents the radius of snow particles as simple shapes, such as spheres or rods, with similar optical properties as the actual snow particles. These simplified shapes have radii that are similar to those of the branch width of the actual snow grains (Warren, 1982). Snow grains experience rapid



changes in size and morphology after snowfall, notably once the snowpack is warmed to its melting point. In dry snow, the gradual coarsening of individual snow grains decreases albedo and enhances the warming process (Picard et al., 2012). The presence of liquid water or light absorbing particles (LAPs) also accelerates snow metamorphism, leading to positive feedbacks between grain growth and snow albedo (Skiles et al., 2017; Tuzet et al., 2017). Grain size has a limited impact on albedo in the

visible spectrum, but albedo in the near-infrared (NIR) varies inversely with optical grain size (Wiscombe and Warren, 1980). Thus, snow grain size is a vital component of snowpack modeling.

The importance of snow grain size has led to the development of retrieval algorithms from spectral reflectance and spectral imaging. Qualitative classifications of grain size were presented by Dozier and Marks (1987), who used Landsat Thematic Mapper data to sort snow into coarseness regimes. Nolin and Dozier (1993) introduced the first quantitative approach using ra-

diance data from a single spectral band of the Airborne Visible/Infrared Imaging Spectrometer (AVIRIS). A more sophisticated technique was developed by Nolin and Dozier (2000) that utilized multiple AVIRIS bands centered at the ice absorption feature at 1.03 $\mu$m to generate an inversion model. A suite of studies has applied the Nolin and Dozier method (henceforth referred to as ND2000) since its inception through contact and imaging spectroscopy (Donahue et al., 2020; Dozier et al., 2009; Painter et al., 2007, 2013; Seidel et al., 2016; Skiles et al., 2017).

Studies using the ND2000 retrieval algorithm often rely on three assumptions: (i) individual ice particles are treated as spheres (Donahue et al., 2020; Painter et al., 2007), (ii) the snowpack impurity content is unlikely to impact the retrieval (Seidel et al., 2016), and (iii) illumination and viewing angles need to be considered (Donahue et al., 2020; Nolin and Dozier, 2000). Previous studies established that the spherical particle assumption works for bulk albedo calculations (Grenfell et al., 2005; Grenfell and Warren, 1999; Neshyba et al., 2003), but it overestimates the scattering asymmetry parameter ($g$), leading

to inaccuracies in snow radiative transfer models that assume spheres (Dang et al., 2016; Kokhanovsky and Zege, 2004; Libois et al., 2013). Furthermore, if dust content is sufficiently high, the dust may increase albedo at near-infrared wavelengths and interfere with grain size retrievals (Nolin and Dozier, 2000).

If a surface is a diffuse reflector (i.e. reflects light in all directions equally), it is known as a Lambertian surface because reflectance can be described by Lambert's cosine law. Snow can be assumed to be a Lambertian surface when the solar illumi-

nation angle is near-zenith over flat surfaces. However, snow reflectance near 1.03 $\mu$m (in the NIR) is anisotropic, preferentially scattering light in the forward direction at higher illumination angles (Dumont et al., 2010; Li, 2007; Picard et al., 2020). Because snow is typically found at high latitudes or on sloped terrain, the illumination and viewing angles must be considered when retrieving snow properties from spectral reflectance. Therefore, a quantitative assessment of the potential impacts of solar geometry and snowpack state on the accuracy of the ND2000 algorithm is needed.

In this study, we used radiative transfer models to examine the sensitivity of snow grain size retrievals to four perturbations: dust content, anisotropic reflectance, solar zenith angle, and ice particle asphericity. The paper is organized as follows: we first describe the methods we used to assess grain size sensitivity, and the radiative transfer models used for this purpose. Section 3 shows the results of our sensitivity tests and discusses the implications for actual grain size retrievals. Section 4 concludes the paper with recommendations for future work.



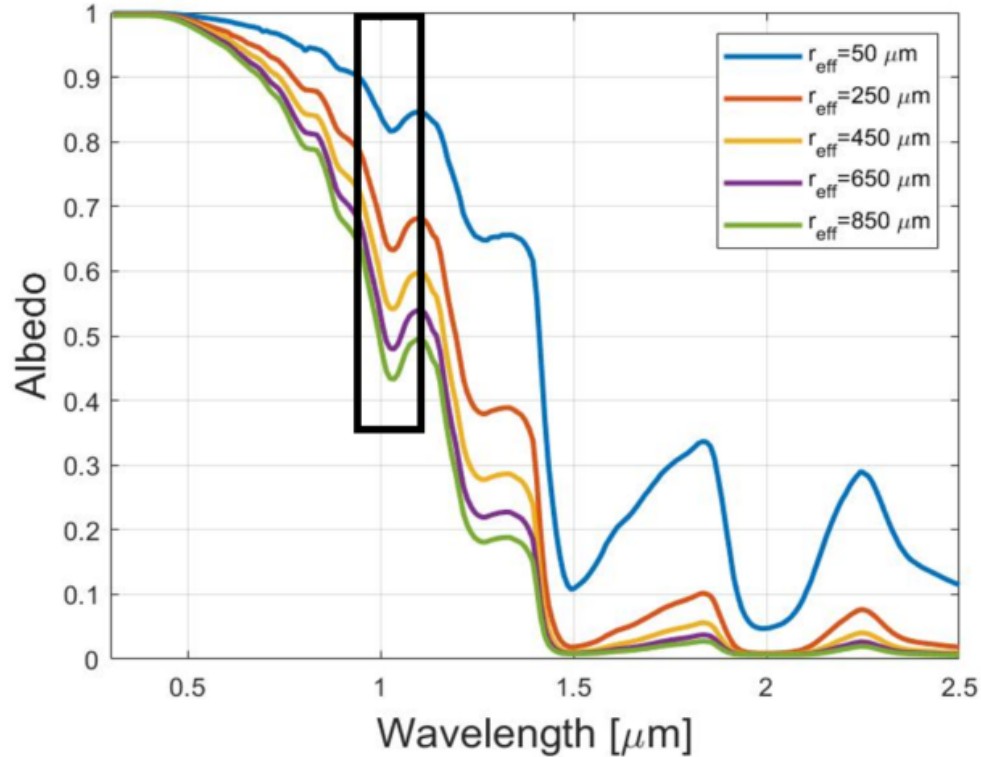

**Figure 1.** The spectral dependence of snow directional-hemispherical albedo as a function of effective snow grain size, as derived by SNICAR. The reflectance curves were modeled assuming spherical ice particles and a solar zenith angle of $60°$. The black box highlights the domain used for grain size retrievals.

## 2 Methods

### 2.1 General description of grain size retrievals

The ND2000 technique estimates snow grain size using directional reflectance at the ice absorption feature centered at 1.03 $\mu$m. Reflectance in this feature decreases as snow grain size increases (Fig. 1), leading to an increase in depth of the absorption feature. This quantity, also known as band depth, is the difference between reflectance without the absorption feature (continuum reflectance) and observed reflectance. Preliminary research by Nolin and Dozier (1993) demonstrated that a single band depth within the ice absorption feature could be used to derive snow grain size, though the method was subject to sensor noise and uncertainties due to local topography. Nolin and Dozier (2000) accounted for the latter issue by scaling band depth relative to the continuum reflectance, which is linearly interpolated between 0.95 $\mu$m and 1.09 $\mu$m. This scaling generates a continuum-



removed spectrum that is independent on the magnitude of reflectance. The former issue was accounted for by instead deriving a scaled band area:

$$A_{b,s} = \int\limits_{0.95\mu m}^{1.09\mu m} \frac{R_c - R_b}{R_c}\, d\lambda \tag{1}$$

where $R_b$ is the spectral reflectance and $R_c$ is the continuum reflectance. The the integrand of Eq. 1 is the scaled band depth
at each wavelength within the absorption feature.

Band area is computed from an observation of spectral reflectance and best matched to a band area within a lookup table or to a calibration curve of modeled band areas. Previous studies derived lookup tables of scaled band area using the Discrete-Ordinates Radiative Transfer (DISORT) model (Stamnes et al., 1988). Here, we instead derived calibration curves using the SNICAR model (Flanner et al., 2007) and a Monte Carlo photon tracking model (Schneider et al., 2019) to derive hemispherical
albedo and directional reflectance, respectively. The reflectances were derived at 14 wavelengths between 0.95 $\mu$m and 1.09 $\mu$m for both models, from which a band area was calculated using Eq. 1. We used polynomial regression to generate the calibration curves to relate grain size to band area for a given set of solar zenith angles or snowpack perturbations. We also used SNICAR and the Monte Carlo model to produce synthetic observations of hyperspectral snow albedo to assess the influence of snowpack variables. This allowed us to evaluate how these features affect grain size retrievals when they are or are not considered in the
creation of the retrieval function.

We quantified the bias of simulated grain size retrievals ($\Delta r$) as the difference between synthetic observations of grain size ($r'$) and the true grain size ($r_0$):

$$\Delta r = r' - r_0 \tag{2}$$

If $\Delta r$ is negative, then the retrieved grain size is smaller than the actual grain size. Conversely, a positive $\Delta r$ implies a larger
retrieved grain size than the actual snow grain size.

## 2.2 Simulated snowpack perturbations

### 2.2.1 SNICAR

The Snow, Ice, and Aerosol Radiative (SNICAR) model incorporates a two-stream radiative transfer solution over a single-layer, semi-infinite snowpack to simulate spectral reflectance at 10 nm resolution. We used version 3 of the model, also known
as SNICAR-ADv3 (Flanner et al., 2021), which incorporates the delta-Eddington approximation and an adding-doubling (AD) technique (Dang et al., 2019). By default, the model handles spheres and multiple aspherical particle shapes (He et al., 2017). Solar zenith angle and snow impurity content serve as inputs to the model, allowing for estimates of spectral albedo given solar geometry or perturbed snowpack conditions. The SNICAR model is less computationally expensive than the Monte Carlo model, so we used it for case studies not focused on anisotropic reflectance, which is not resolved by SNICAR and other




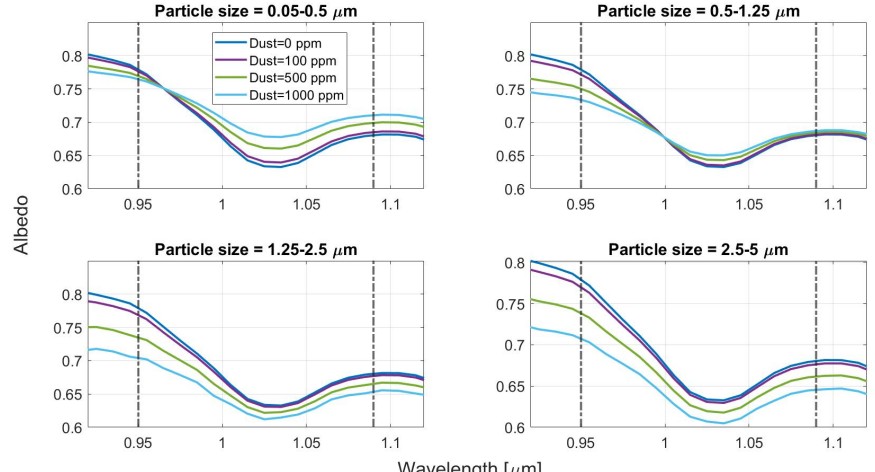

**Figure 2.** Spectral albedo of snow derived from SNICAR as a function of Saharan dust content at near-infrared wavelengths, given four particle size distributions and $r_{eff} = 250\ \mu$m. The dashed lines indicate the bounds for band area calculations. Dust concentrations at 1 ppm and 10 ppm were identical to clean snow with the given configurations and were thus omitted.

two-stream models. As a baseline, we generated a calibration curve for snow grain sizes of 50-1000 $\mu$m at 50 $\mu$m intervals, assuming a solar zenith angle of $60°$, spherical ice particles, and zero impurity content. We compared the grain size retrievals of perturbed snowpacks to this baseline when SNICAR was used.

### 2.2.2 Snowpack perturbations

We assessed each snowpack variable independently to highlight their individual effects on grain size retrievals. We assumed direct sunlight for all simulations. Spectra were modeled for a range of solar zenith angles ($\theta_0$), snow grain shapes, and LAP concentrations. For our analysis on solar zenith angle, we considered angles at near-horizon or near-zenith unlikely for most grain size retrieval conditions, so we restricted our simulations to $\mu_0 = cos\,\theta_0 = [0.3, 0.4, 0.5, 0.6, 0.7]$. To examine the influence of ice particle asphericity, we used the available ice particle shapes in SNICAR-ADv3: spheroids, hexagonal plates,

and Koch snowflakes (i.e. aspherical particles with a fractal orientation). These particle shapes are simulated by assigning a radius to spherical particles of equivalent specific surface area. The Mie properties used for spherical particles produce values of $g = 0.88$-$0.90$ over the part of the spectrum used for retrievals, compared to the $g$ values of 0.85, 0.8, and 0.75 for spheroids, hexagonal plates, and Koch snowflakes, respectively. Grain size retrieval errors are calculated relative to calibration functions that do not account for variations in solar zenith angle or ice particle asphericity.

We analyzed retrieval errors of contaminated snow with four different types of light-absorbing particles: Saharan dust (Balkanski et al., 2007), San Juan dust (Skiles et al., 2017), Greenland dust (Polashenski et al., 2015), and black carbon. The dust species were assessed at four size distributions: 0.05-0.5 $\mu$m, 0.5-1.25 $\mu$m, 1.5-2.5 $\mu$m, and 2.5-5.0 $\mu$m, whereas black carbon (BC) was analyzed for only one size distribution. The particle optical properties of these species are described in (Flanner et al.,



2021). We selected dust concentrations based on their impact on near-infrared reflectance. Dust only affects NIR albedo when its content is high (~100 ppm), otherwise changes are restricted to the visible spectrum (Fig. 2). We therefore examined five concentrations for dust: 1 ppm, 10 ppm, 100 ppm, 500 ppm, and 1000 ppm. To account for its greater impacts on albedo, BC concentrations are given in amounts of parts-per-*billion* (ppb) rather than parts-per-million. Grain size retrieval errors are then
calculated via calibration functions that assume pure snow.

### 2.3 Anisotropic reflectance modeling

#### 2.3.1 Monte Carlo model

To analyze the importance of anisotropic reflectance, we used a Monte Carlo model originally developed by Schneider et al. (2019), which calculates azimuthally-averaged bidirectional reflectance factors (BRF) for idealized snowpack configurations.
In the model, photons propagate through a highly scattering semi-infinite medium of ice particles until they are terminated (absorbed) or escape (reflected). Ice particles are assumed to have scattering phase functions that follow the Henyey-Greenstein phase function (van de Hulst, 1968), with scattering asymmetry parameters derived from the full scattering phase functions presented by Yang et al. (2013), who assume randomly-oriented particles. Schneider et al. (2019) showed that the Henyey-Greenstein function produces similar snow reflectance patterns as the full phase function, but with greatly reduced computa-
tional cost. Given solar zenith angle ($\theta_0$) and reflected/viewing angle ($\theta_v$), the BRF is calculated using

$$BRF(\theta_0; \theta_v) = \frac{\int_0^{2\pi} \Phi_r(\theta_v, \phi_v) \, d\phi_v}{2\Phi_i(\theta_0) sin(\theta_v) cos(\theta_v)} \tag{3}$$

where $\Phi_i(\theta_0)$ is the incident photon flux from solar angle $\theta_0$ and $\Phi_r(\theta_v, \phi_v)$ is the photon flux received by a sensor at azimuth angle $\phi_v$ and elevation angle $\theta_v$, assuming that 0° is nadir. The azimuthally-averaged BRF is defined using Lambert's cosine law, so the averaging requires a weighting factor of $\omega(\theta_v) = (2 \sin \theta_v \cos \theta_v)^{-1}$. In this form, the BRF represents a ratio
between actual reflectance and reflectance over a Lambertian surface with equal albedo.

#### 2.3.2 Anisotropy configurations

We performed Monte Carlo simulations with one million photons at each wavelength, which offered a compromise between reduced noise and increased computational expense. Photons that escaped from the top of the snowpack were used to estimate BRF using Eq. 3. The calculated reflectances were distributed among 30 bins of zenith angle at 3°resolution for five snow
grain sizes: 50 $\mu$m, 250 $\mu$m, 450 $\mu$m, 650 $\mu$m, and 850 $\mu$m. Although using fewer grain sizes reduces the resolution of the calibration curve, we deemed it a necessary step to reduce computational cost. For each grain size, BRF was estimated given $\theta_0 = 0°$, 15°, 30°, 45°, 60°, and 75°.

Spectral reflectance measurements are often made at near-nadir viewing angles (Gao et al., 1993), so we tested for anisotropy at $\theta_v = 0$-15°, which we henceforth refer to as the bidirectional reflectance or BRF. Directional reflectance calculated with
Monte Carlo techniques is subject to random photon noise, so we applied a second-order polynomial fit to the spectral BRF

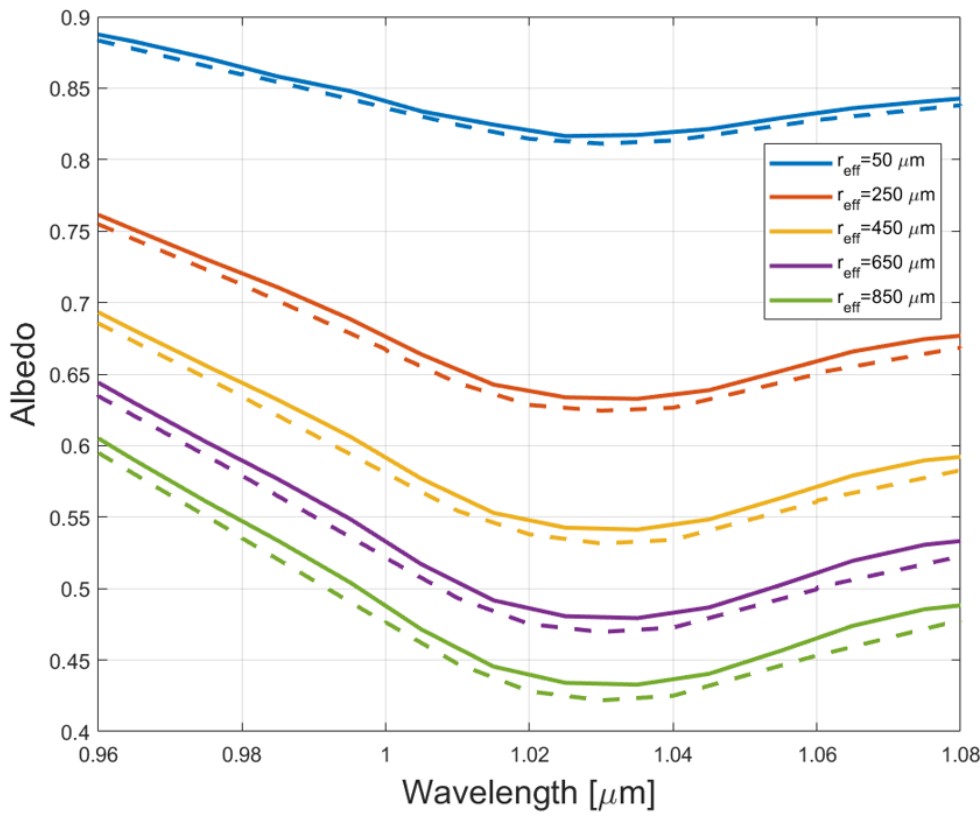

**Figure 3.** Spectral (directional-hemispherical) reflectance of snow without impurities calculated using the SNICAR model (solid) and the Monte Carlo model (dashed). The reflectances were derived for multiple grain sizes using $\theta_0 = 60°$.

output to smooth out noisy features. Preliminary analysis shows that hemispherical albedo derived from the Monte Carlo model agrees very closely with that of SNICAR at the given snow grain sizes (Fig. 3).

For tests on the influence on both solar zenith angle and snow grain shape, we configured the Monte Carlo model to generate BRF estimates for three particle configurations: droxtals, hexagonal plate aggregates, and solid column aggregates. The droxtals and plate aggregates are nearly equivalent to spheroids and hexagonal plates in SNICAR-ADv3, respectively, whereas column aggregates have an asymmetry factor slightly larger than Koch snowflakes. The scattering phase functions of these particles assume that the ice particles are randomly oriented within the snowpack. The aspherical particle tests were performed given $r_{eff} = 250$ $\mu$m and the solar zenith angles given above.





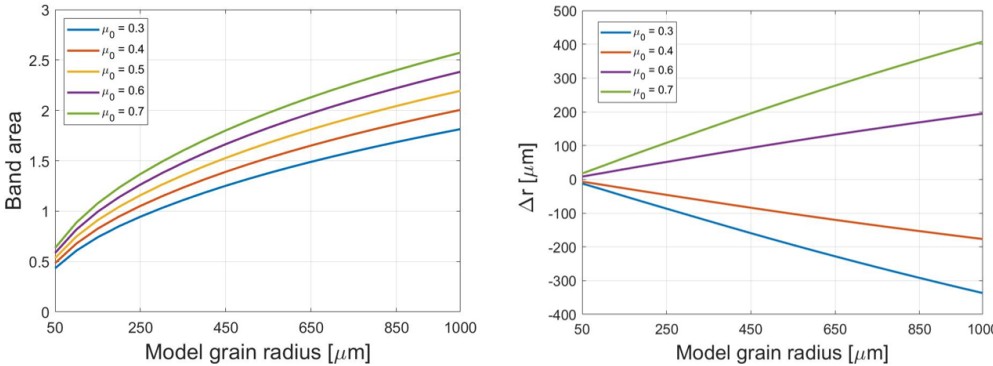

**Figure 4.** Band area as a function of grain size and solar zenith angle (a) and the corresponding grain size biases (b). The term "$\mu_0$" refers to the cosine of the solar zenith angle. Biases are computed relative to the baseline calibration function assuming $\theta_0 = 60°$ ($\mu_0 = 0.5$).

Previous studies by Nolin and Dozier (2000) and Donahue et al. (2020) established that scaling band area relative to a continuum removes its dependence on the magnitude of reflectance, thereby reducing the impact of illumination angle variability. To validate this point, we performed additional anisotropy tests using unscaled band area $A_{b,u}$, which is given by

$$A_{b,u} = \int\limits_{0.95\mu m}^{1.09\mu m} R_c - R_b \, d\lambda \tag{4}$$

For both scaled and unscaled band area, we performed three tests dependent on the reflectance quantities used for lookup table generation and for simulated retrievals. The first test applied a calibration curve derived from hemispheric reflectance and also assumed that hemispheric reflectance (i.e., albedo) is also the measured snow reflectance quantity. This configuration is equivalent to the snow grain size retrievals performed with SNICAR. The second test instead used BRF for the measured reflectance and left the calibration curve unchanged. Snow grain size retrievals performed with this configuration demonstrated
the effects of anisotropy without corrections. The final test utilized BRF for both the calibration curve and the measured reflectance and thus served as a correction for anisotropy.

## 3 Results and Discussion

### 3.1 Solar zenith angle

For this analysis, the band areas used to create both the calibration curves and the modeled retrievals and the corresponding
grain size errors were derived using hemispheric albedo from SNICAR. Our results for the solar zenith angle sensitivity study are given in Fig. 4. Band area changes proportionally to the cosine of the illumination angle ($\mu_0$), as reported by Donahue et al. (2020). Band area is most sensitive to $\mu_0$ when the Sun approaches the horizon ($\mu_0 = 0.3$ case), where reflectance is higher and less wavelength-dependent. When $\theta_0$ is close to our calibration baseline of $60°$, biases remain reasonably low for all but the





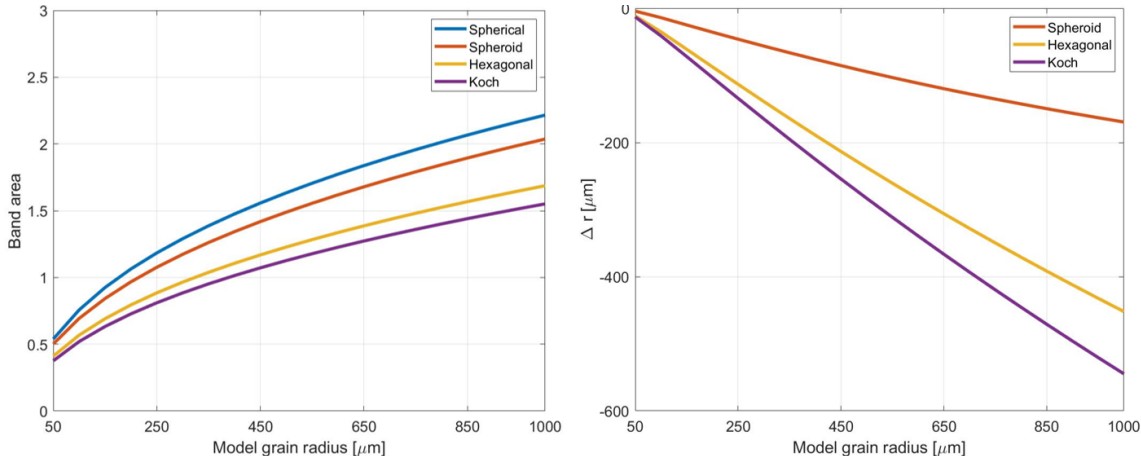

**Figure 5.** Same as Fig. 4, but for changes in ice particle shape. Estimated biases are relative to spherical ice particles.

largest snow grain sizes ($\geq$500 $\mu$m). Errors may exceed 300 $\mu$m as $\theta_0$ deviates from the baseline, but otherwise remain within 100 $\mu$m.

When solar zenith angle changes, the likelihood of photon absorption within the snowpack also changes. Incident sunlight penetrates into a snowpack more effectively as $\theta_0$ approaches zenith, allowing for more opportunities for absorption or multiple
scattering and decreasing spectral albedo. To the retrieval algorithm, this "darker" surface corresponds to a deeper absorption feature, increasing scaled band area and apparent snow grain size. The opposite is true when $\theta_0$ approaches the horizon. The biases described above illustrate the importance of incorporating solar zenith angle into the retrieval of grain size when applying the ND2000 algorithm.

### 3.2  Ice particle asphericity

The results for our sensitivity study on ice particle shape (Fig. 5) show a significant increase in bias when particle shape deviates from spherical particles and when spherical particles are assumed in the creation of the calibration function. Differences in band area are non-negligible between spherical and hexagonal and Koch particles. The band area decreases notably between spheroids and hexagonal plates, but the difference between plates and Koch snowflakes is smaller. At 1000 $\mu$m, the difference in band area between spheroids and hexagonal plates is 0.35, compared to a difference of 0.14 between plates and Koch
snowflakes. The biases are large for model grain sizes of 500 $\mu$m or higher for all aspherical particles, with a maximum bias of 545 $\mu$m for Koch snowflakes. However, bias appears to be significant only when model grain size is greater than 200 $\mu$m. At smaller grain sizes, retrieval errors are less sensitive to particle asphericity.

When the asymmetry parameter changes value, it affects reflectance in ways similar to solar zenith angle. Spherical particles scatter visible and NIR radiation in the forward direction more strongly than other particle shapes, leading to a lower observed
albedo. As with near-nadir illumination angles, the lower albedo is interpreted as a larger grain size by the algorithm. If the true particle shape is sufficiently non-spherical, the albedo will increase in the ice absorption feature and reduce retrieved





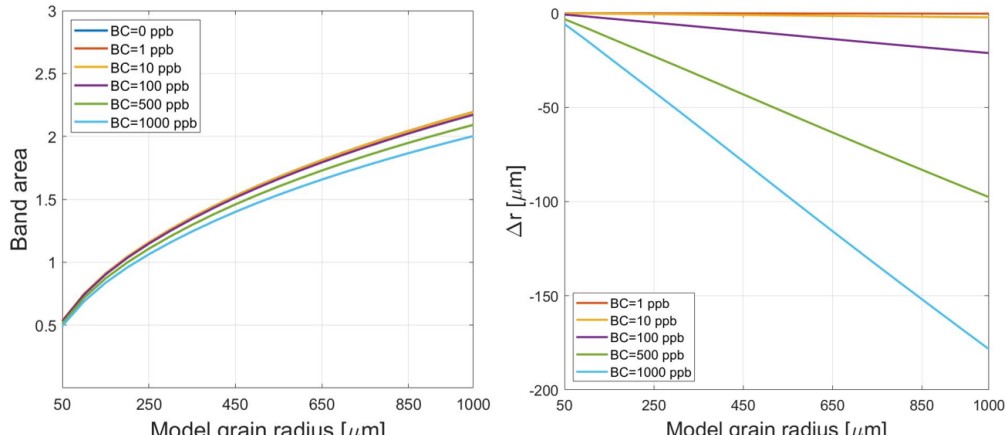

**Figure 6.** Same as Figs. 4 and 5, but for changes in black carbon content. Grain size errors are calculated from calibration curves assuming no impurity content.

snow grain size. In nature, freshly fallen snow generally begins as small, non-spherical particles before aggregating into larger spheroids (Sturm and Benson, 1997). The spherical particle assumption is therefore most valid for aged snow, whereas a fresh snowpack may be less predictable due to the larger variety in grain shapes.

### 3.3 Black carbon and dust

Relative to a clean snow case, we found that a snowpack requires a high concentration of black carbon to impact the 1.03 $\mu$m ice absorption feature. Relative to the baseline with no impurity content, calibration curves with concentrations below 500 ppb show minimal effect on band area or grain size retrievals (Fig. 6). Band area decreases more efficiently when black carbon exceeds 500 ppb, implying that it begins to supplant ice absorption at these levels. However, this circumstance only occurs for coarse-grained snow. The maximum observed bias is 178 $\mu$m at 1000 ppb of BC, but bias decreases to below 100 $\mu$m or less

for grain sizes smaller than 500 $\mu$m.

    The three dust species show similar trends in band area and grain size bias for all concentrations and particle size distributions (PSD). The results in Figs. 7 and 8 therefore apply to all species, despite slight differences in absorptivity. For all PSD, band area is unperturbed when dust content is 10 ppm or less, and larger PSD show further insensitivity at 100 ppm. The differences become more significant at larger concentrations, namely for large snow grain sizes and small PSD. Retrieval biases become

substantial in extreme situations, with 1000 pm of dust producing an error of 829 $\mu$m for true snow grain size $r_0 = 1000$ $\mu$m and dust particle radius = 0.05-0.5 $\mu$m. When dust content is $\geq$500 ppm, biases are significant ($\Delta r \approx 200$ $\mu$m) even at small grain sizes. The bias diminishes with larger particles, though it still exceeds 300 $\mu$m when 1000 ppm of dust is present. The decrease in band area with dust also appears to saturate at large concentrations, as the ice absorption feature becomes obscured.

    The impact of high dust content on dampening of the absorption feature was recognized by Skiles et al. (2017), but it was

not quantitatively investigated. Both Seidel et al. (2016) and Skiles and Painter (2019) also postulated that dust influences snow





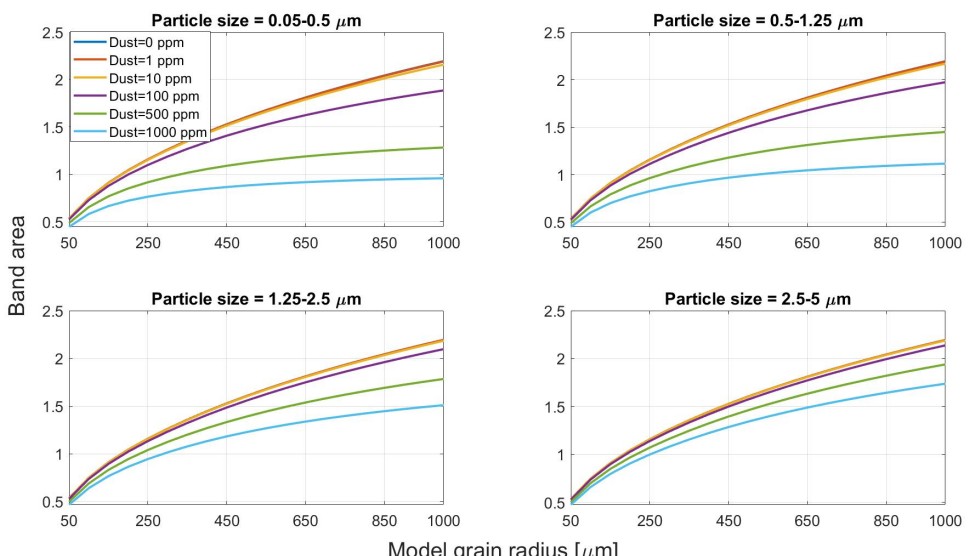

**Figure 7.** Band area sensitivity to modeled snow grain size and San Juan dust content. Sensitivities are given for the four particle size distributions.

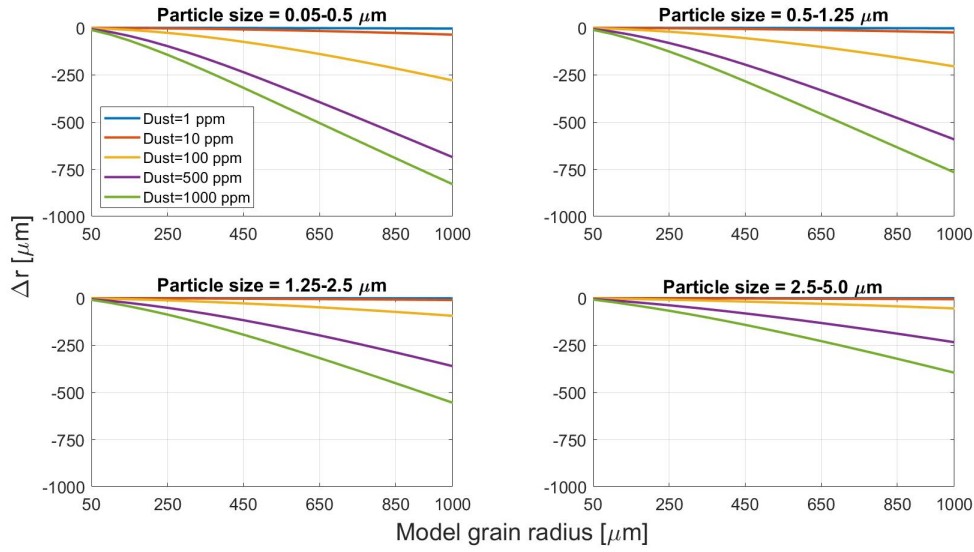

**Figure 8.** Grain size retrieval biases for San Juan dust of four size distributions. Biases are relative to a clean snow case (i.e. Dust=0 ppm).



grain size through enhanced metamorphic processes, an effect verified by Schneider et al. (2019) in near-freezing, clear-sky conditions. The results here suggest that dust also "masks" the ice absorption feature by reducing albedo at the left shoulder. Dust with small PSD also appears to increase albedo at 1.03 $\mu$m, thereby reducing band area further. Although there is uncertainty in the refractive indices of dust and black carbon, particularly in the near-IR, we expect any impurity in sufficient

quantity to flatten the 1.03 $\mu$m ice absorption feature because this feature is unique to $H_2O$. The measured band area of a dirty snowpack will be small, leading to a strong negative bias in the retrieved grain size. The impacts are most severe for small particle sizes, which cause greater extinction per unit mass of impurity than larger particles. In worst case scenarios (e.g., Fig. 8a), a retrieval performed over a snowpack with $r_0 = 1000$ $\mu$m would return a grain size of less than 200 $\mu$m. Prior knowledge of snowpack impurity content is therefore essential to avoid biases when measuring dirty, coarse-grained snow.

On a per-mass basis, black carbon exhibits a stronger influence on NIR reflectance than dust. A snowpack with 1 ppm of dust shows no bias in grain size, whereas this concentration of black carbon affects retrievals by 100 $\mu$m or more when $r_0 \geq 500$ $\mu$m. However, such concentrations of black carbon are uncommon in nature, only occurring near heavy BC sources (Flanner et al., 2007). Natural BC concentrations are typically much less than 100 ppb, which are shown in Fig. 6 to have minimal impact on grain size retrievals. Episodic dust deposits are more likely to generate significant biases at regional scales, as evidenced by the

8000 ppm of dust observed by Skiles and Painter (2017) in the San Juan Mountains. Although dust deposited on Greenland has the theoretical potential to induce errors, significant dust or black carbon deposits are rare over the ice sheet (Polashenski et al., 2015; Ward et al., 2018), so the risk is reduced relative to mid-latitude locations. However, parts of the Greenland ablation zone are very dark due to algae and other organic matter (Cook et al., 2020), so similar impurity-related biases could exist in these regions.

## 3.4  Anisotropic reflectance

The angular distribution of BRF at 1.035 $\mu$m is shown in Fig. 9 for six illumination angles: 0°, 15°, 30°, 45°, 60°, and 75°. When $\theta_0 \leq 30°$, the BRFs are effectively isotropic for viewing angles up to 45° and small snow grain sizes, and the magnitude of reflectance decreases as $\theta_v$ approaches the horizon. The BRF distribution is more uniform at larger snow grain sizes, with BRF reductions occurring at $\theta_v \geq 60°$. Anisotropy becomes more pronounced at larger solar zenith angles. Reflectance

decreases at near-zenith angles and peaks near the horizon, meaning that forward scattering peaks at large angles due to a shallower penetration depth. The BRF is nearly 2.0 when $\theta_0 = 75°$ and $\theta_v \geq 75°$, suggesting that reflectance substantially exceeds that of a white (or lossless) Lambertian reflector at these angles. We also note that the BRF is averaged across all azimuthal angles, so the maximum BRF will be much higher in the plane of illumination.

Similar patterns in BRF are seen among aspherical particles (Fig. 14), in that reflectance is nearly isotropic at near-zenith $\theta_0$

and highly anisotropic at $\theta_0 = 60°$ and 75°. Column and droxtal shapes show higher reflectance at nearly all viewing angles, as suggested by the results in Fig. 5. The exception is at $\theta_0 = 75°$ and $\theta_v \geq 70°$, where spherical particles exhibit larger reflectance than other particles. This change is likely caused by the larger scattering asymmetry parameter of spherical ice particles. It is unclear why the reflectance for hexagonal plates is comparable to that of spherical particles. Other than these differences, there are negligible changes in the angular distribution of BRF between spherical and aspherical particles.





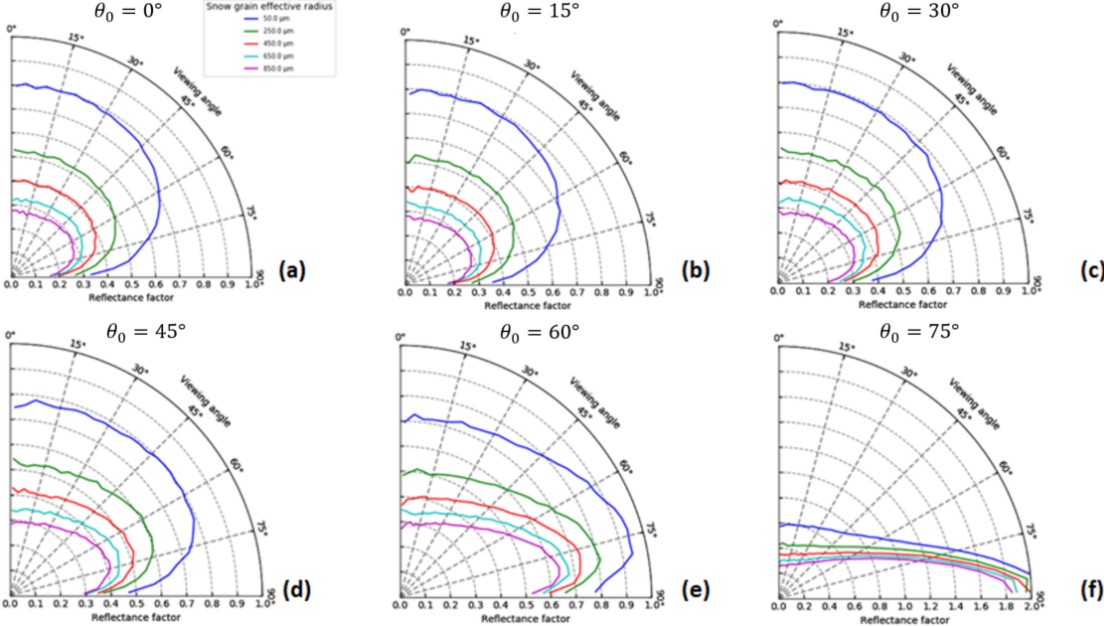

**Figure 9.** Polar plots of azimuthally-averaged bidirectional reflectance factors (BRF) of modeled snowpacks with various snow grain radii, six illumination angles, and $\lambda = 1.035\ \mu$m. The radial dimension for (f) has been extended to capture the full extent of the localized BRF.

Spectral reflectance curves derived from hemispheric reflectance and BRF (Fig. 10) exhibit similar shapes at the ice absorption feature, despite differences in reflectivity. The BRF exceeds the hemispheric reflectance when illumination angle is near-zenith, as seen in Figs. 10a and 10b, and there is little change in reflectivity between the two angles. At $\theta_0 = 30°$, the reflectance curves are nearly identical, with slight overestimates in the hemispheric albedo at 0.95 $\mu$m and underestimates at 1-1.07 $\mu$m. The continuum reflectance (the dotted lines in Fig. 10) of BRF is higher for 0.95-1.07 $\mu$m before converging to the hemispheric mean at the right shoulder of the absorption feature.

Figure 11 shows how unscaled band area (Eq. 4) and scaled band area (Eq. 1) differ for changing solar zenith angle. The unscaled band area at large illumination angles is similar between hemispheric reflectance and BRF, despite differences in absolute reflectance (Fig. 10). Agreement in $A_{b,u}$ between hemispheric reflectance and BRF decreases as $\theta_0$ decreases, with the most significant differences occurring between 250 $\mu$m and 450 $\mu$m. However, Fig. 11 also demonstrates that agreement in $A_{b,s}$ improves when $\theta_0 < 45°$, with RMSE decreasing from 0.79 at 75° to 0.29 at 0°. The disagreement in $A_{b,s}$ between the BRF and hemispheric reflectance worsens at large illumination angles, given that the near-zenith BRF decreases significantly relative to the hemispheric mean as anisotropy increases (Figs. 9d-f and 10d-f). There is little change in agreement between 0° and 30°, which is expected given the results from Figs. 9 and 10.

The effects of anisotropy on grain size retrievals are given in Table 1 for the six illumination angles. The errors shown in Columns 2 and 3 are with calibration curves derived from hemispheric albedo, whereas Column 4 uses a calibration function



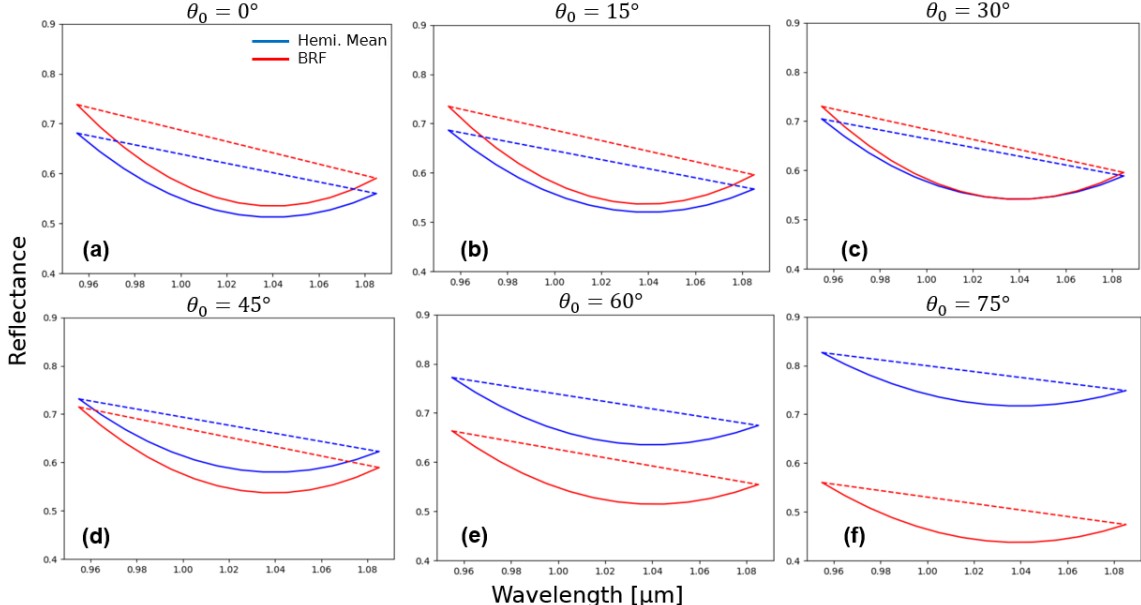

**Figure 10.** Spectral reflectance integrated across a hemisphere ("Hemi. Mean", blue) and for the average BRF received at 0-15°("BRF", red) for $r_{eff} = 250$ $\mu$m at the prescribed illumination angles. The dashed lines represent continuum reflectance for the corresponding spectral curves.

derived from directional reflectance. The RMSE range for the baseline simulation is 1.6-4.8 $\mu$m for scaled band area (Column 1), implying that uncertainties inherent to the ND2000 method are small. The unscaled band area shows greater uncertainty at all angles but is smallest when $\theta_0$ is large. When the modeled retrieval is of BRF but the calibration is derived from hemispheric albedo (Table 1, Column 3), RMSE in grain size remains within 200 $\mu$m when reflectance is nearly isotropic. The errors

increase exponentially as anisotropy becomes more significant, with $\Delta r$ exceeding 1000 $\mu$m at illumination angle 75° for grain sizes ≥650 $\mu$m (Fig. 12). When the calibration curve was derived using BRF, consistent with the synthetic observation, errors dropped significantly across all illumination angles. Figure 13 shows that the maximum RMSE among the corrected retrievals is 17 $\mu$m at 75°, corresponding with a maximum $\Delta r$ of 23.2 $\mu$m at input grain size 250 $\mu$m.

The sensitivity of band area to anisotropic reflectance depends on the usage of continuum scaling. Band area without scaling

performs best at high solar zenith angles, where retrieval errors resulting from the Lambertian assumption remain low even when no correction is applied. Reflectance spectra exhibit fewer differences in curve shape, thereby reducing retrieval errors. In contrast, reflectance at smaller illumination (zenith) angles is nearly isotropic. The hemispheric reflectance and BRF generally agree to within 0.02, but because $A_{b,u}$ is small, it is highly sensitive to differences in BRF and consequently produces significant grain size errors even when the correct retrieval scheme is used. When band area is scaled, grain size retrievals become more

accurate at lower illumination angles. Although small differences exist between hemispheric reflectance and BRF, $A_{b,s}$ is larger



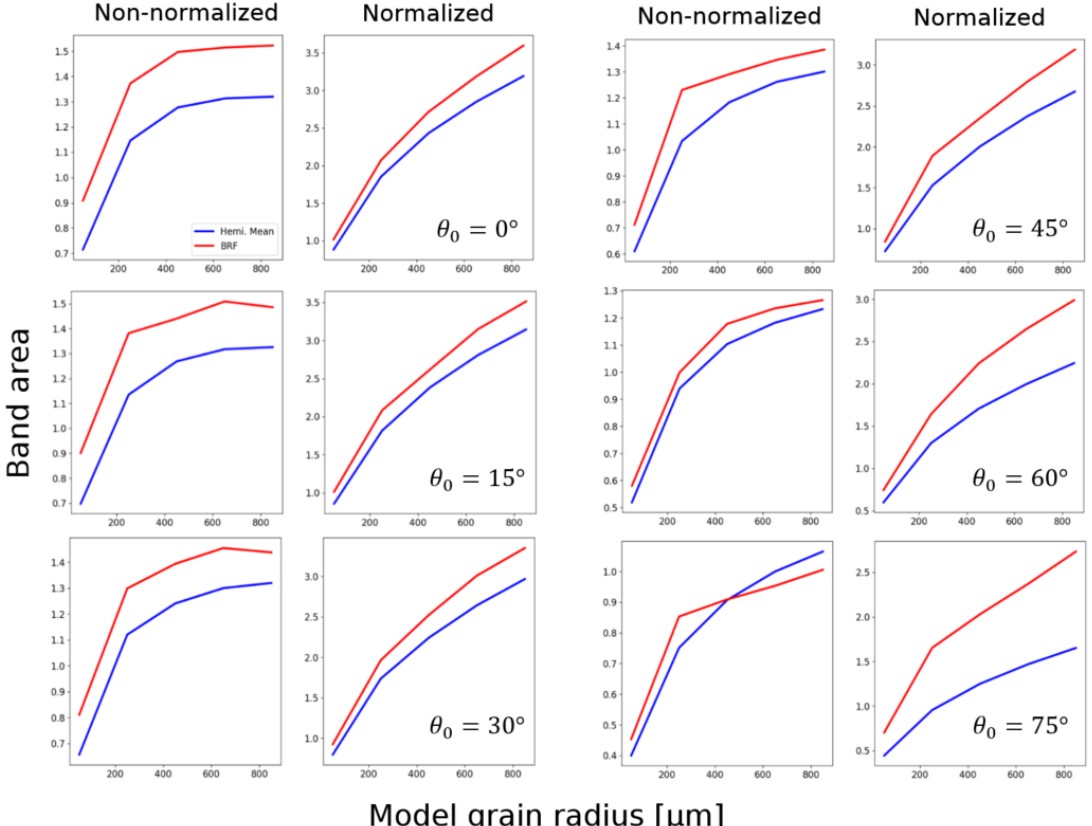

**Figure 11.** Calibration curves for band area vs. model grain radius, derived using the hemispheric reflectance (blue) and BRF (red) curves from Fig. 10. Columns 1 and 3 use band area without continuum scaling (Eq. 4), whereas Columns 2 and 4 are calculated using Eq. 1.

in magnitude than $A_{b.u}$, so it is relatively insensitive to noise in isotropic profiles. Scattered radiation tends more strongly to the horizon as illumination angle increases, leading to the large differences seen in Fig. 11.

For both $A_{b,u}$ and $A_{b,s}$, there is a dependence on illumination angle and model grain size. As $r_0$ increases, the potential bias in a retrieval also increases. Figure 12 shows that errors originating from the Lambertian reflectance assumption at illumination

5   angle $0°$ start at 14.3 $\mu$m before gradually increasing to a peak of 260.5 $\mu$m at large grain sizes. Errors remain within 75 $\mu$m when $r_0 = 50$ $\mu$m at all illumination angles, but increase exponentially with grain size and solar zenith angle. The increase in error is greatest when solar zenith angle increases from $60°$ to $75°$, indicating a significant change in the directionality of reflectance. Biases also increase significantly between grain sizes at $\theta_0 = 75°$. When directional reflectance is used to generate the calibration curve, biases are reduced drastically (Fig. 13).

10   We attribute the significant errors in $A_{b,s}$ at $\theta_0 = 75°$ to changes in continuum reflectance. Figures 10f and 11f indicate that differences in unscaled band area between hemispheric reflectance and BRF are small at large illumination angles. The lack of disparity in $A_{b,u}$ implies that spectral band depth is nearly equal for hemispheric reflectance and BRF, so it can





| $\theta_0$ | Calib.: Hemi. Mean<br>Retrieval: Hemi. Mean | Model: Hemi. Mean<br>Retrieval: BRF | Model: BRF<br>Retrieval: BRF |
|---|---|---|---|
| *Non-normalized* | | | |
| 0° | 83.9 | 791.7 | 97.2 |
| 15° | 74.3 | 641.5 | 99.6 |
| 30° | 53.8 | 485.4 | 92.7 |
| 45° | 38.5 | 266.7 | 8.8 |
| 60° | 35.0 | 126.5 | 61.2 |
| 75° | 19.1 | 101.9 | 14.1 |
| **Mean** | 50.8 | 402.3 | 62.3 |
| *Normalized* | | | |
| 0° | 4.6 | 158.5 | 2.7 |
| 15° | 4.8 | 149.5 | 9.1 |
| 30° | 1.6 | 170.6 | 6.9 |
| 45° | 2.2 | 240.3 | 11.0 |
| 60° | 4.0 | 464.8 | 7.5 |
| 75° | 2.8 | 1053.0 | 17.0 |
| **Mean** | 3.3 | 372.8 | 9.0 |

**Table 1.** Root mean square errors of retrieved snow grain size using non-normalized band area (top half) and normalized band area (bottom half). In the header, "Calib." refers to the reflectance quantity used to generate the lookup table, whereas "Retrieval" is the type of reflectance assumed to be measured.

be concluded that anisotropy is not significantly impacting the 1.03 $\mu$m ice absorption feature. Instead, a notable decrease in continuum reflectance is observed for BRF at large illumination angles, so $A_{b,s}$ will appear much larger than that of hemispheric reflectance, despite similarities in band depth.

## 4 Conclusions

5   We examined the potential sensitivity of snow grain retrievals that exploit the 1.03 $\mu$m ice absorption feature to assumptions about solar illumination angle, snowpack properties, and anisotropic reflectance. Simulations with the SNICAR model showed that retrieval biases are normally within 100 $\mu$m, but incorrect handling of illumination angle and uncertainty in ice particle shape may lead to maximum errors of 400 $\mu$m and 540 $\mu$m, respectively, when the true grain size is large. Black carbon has relatively minor impacts even at large concentrations ($\Delta r = 178$ $\mu$m at maximum), despite its large influence on visible

10  reflectance. Dust biases can exceed 750 $\mu$m when dust is present in high concentration, so estimations of snow dust content may be needed when attempting to retrieve snow grain size, especially in regions affected by large episodic deposition events.





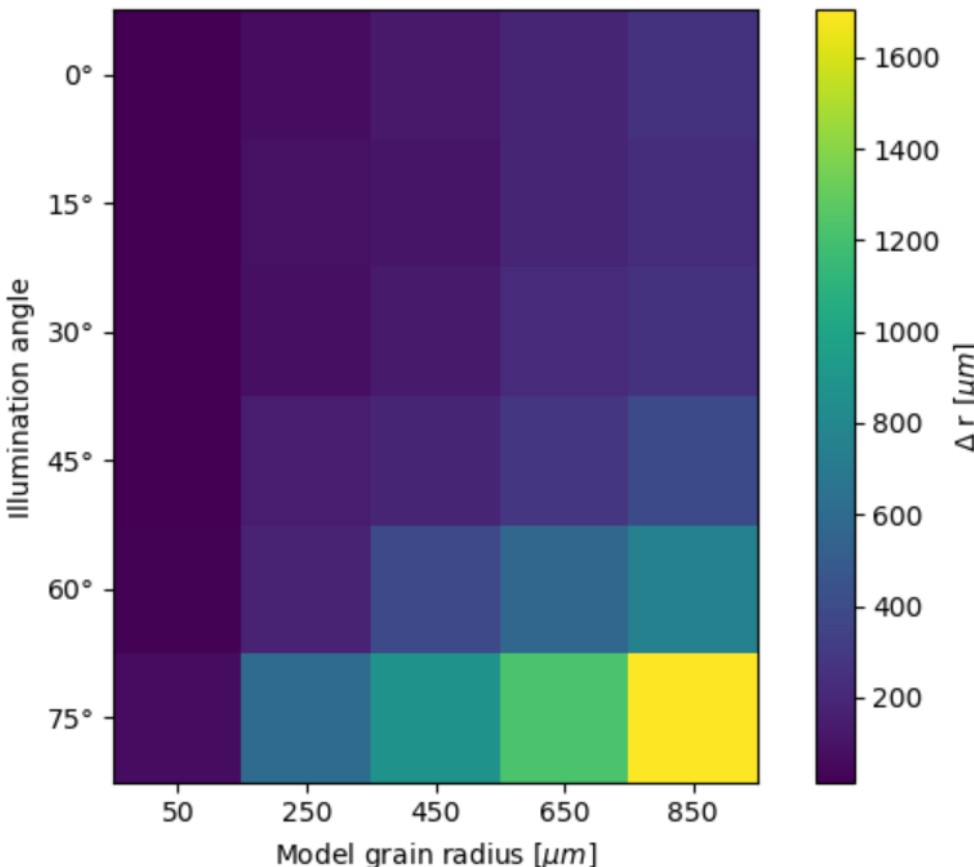

**Figure 12.** Retrieval errors as a function of model grain size and illumination angle, if using normalized band area. The errors assume that the inputs for the calibration curve and the retrieval are hemispheric reflectance and BRF, respectively.

We also assessed the utility of incorporating directional reflectance into the retrieval lookup tables. Our results indicate that hemispheric mean reflectance is an acceptable input into ND2000 at small snow grain sizes and near-zenith illumination angles, where reflected radiation is nearly isotropic. We also observed that changes in ice particle grain shape had little influence on the angular distribution of reflectance. However, larger illumination angles produce up to 1053 $\mu$m even for smaller grain sizes.

5 Our Monte Carlo simulations suggest that band depth is similar between hemispheric reflectance and BRF when anisotropy is significant, but differences in continuum reflectance lead to anomalously large normalized band area for BRF. The retrieval errors decrease substantially to 2.7-17 $\mu$m when directional reflectance is used to generate the lookup table, so it is imperative for future snow grain size retrieval efforts to consider viewing angle, solar geometry, and local topography (Picard et al., 2020).

The results presented here only apply simulated reflectances to evaluate retrieval biases and carry the benefit of having

10 exact knowledge of the "true" grain size. Future studies, however, should explore such retrieval biases with observed hyperspectral data and coincidental in-situ measurements. We do not anticipate significant errors for airborne and field retrievals in



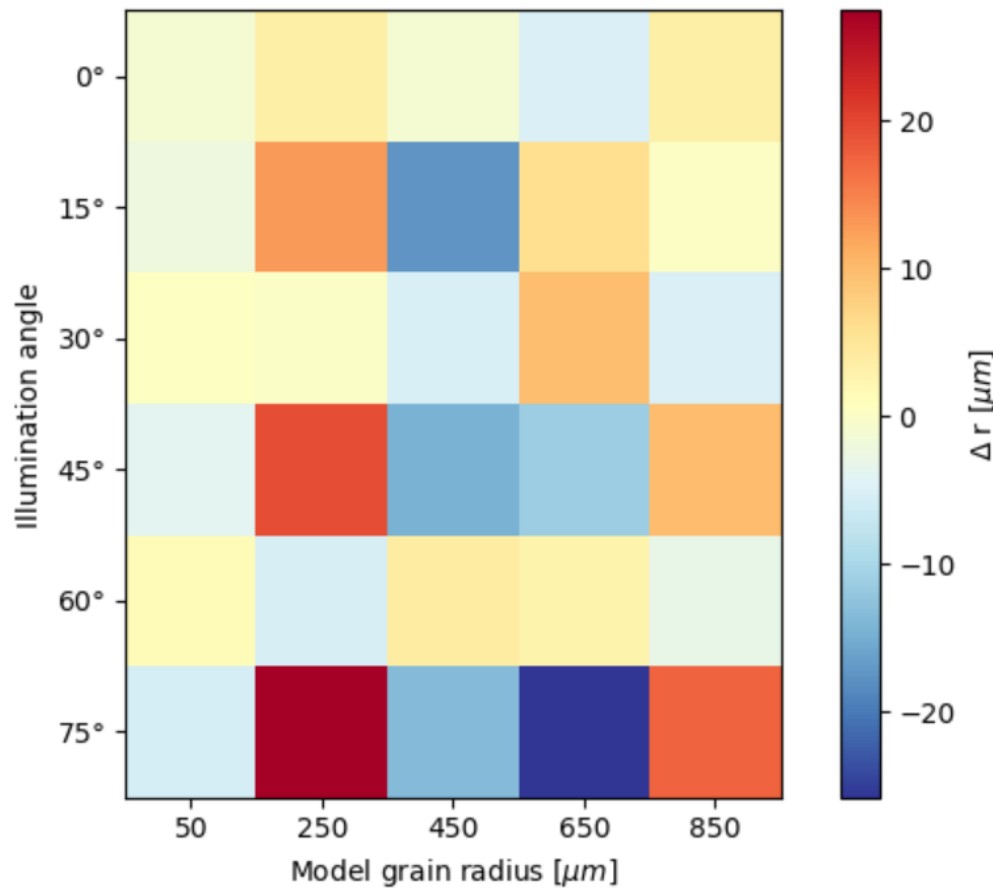

**Figure 13.** Same as Fig. 12 but instead using BRF as input for both the calibration curve and the retrieval.

mid-latitude clean snow, where collections occur during the day at near-zenith solar angles with nadir-viewing sensors. However, we expect that anisotropic reflectance would contribute more significant errors to grain size retrievals over Greenland, where solar zenith angle is high. Future hyperspectral satellite missions, such as Surface, Biology and Geology (SGB) and the Copernicus Hyperspectral Imaging Mission for the Environment (CHIME), may perform acquisitions at different times of day, so anisotropic reflectance will also be a factor in spaceborne retrievals. We considered each snow perturbation separately, so possible relationships and co-dependencies between variables could be assessed in future studies. This is especially true for anisotropic reflectance, where the presence of dust or aspherical particles may further exacerbate retrieval errors.

*Author contributions.* Zachary Fair was responsible for the analysis and writing of the manuscript. Mark Flanner provided guidance on the methods and the SNICAR model. McKenzie Skiles gave insight on airborne grain size retrievals and provided San Juan dust optics. Adam Schneider developed the original Monte Carlo model and gave feedback on its usage.




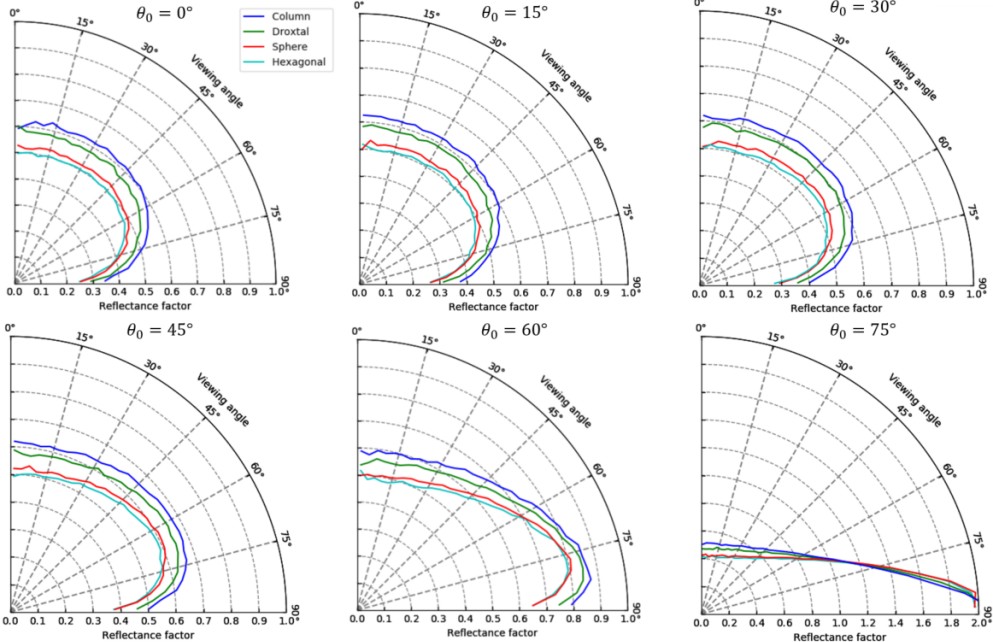

**Figure 14.** Same as Fig. 9, but for various ice particle shapes. The radial dimension for (f) is again extended to account for the high BRF values.

*Competing interests.* The authors declare no competing interests.

*Acknowledgements.* This research was funded by NASA grant 80NSSC20K0062 and the NASA Earth and Space Science Fellowship (grant no. 19-EARTH19R-0047).



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
