# Peer review of "Sensitivity of modeled snow grain size retrievals to solar geometry, snow particle asphericity, and snowpack impurities"

_EGUsphere, 2022_

## Author Comment (AC2)

**Response to Review #2 on the manuscript:**

*Sensitivity of modeled grain size retrievals to solar geometry, snow particle asphericity, and snowpack impurities* by Fair et al.

In this response, the original comments are given in black, the authors' response in blue, and the proposed changes in orange.

Page 3, line 4: "leading to an increase in depth of the absorption feature": is the "depth" defined at the wavelength of 1.03 microns only? Or is it for a wavelength range? Later in this paragraph: "Preliminary research by Nolin and Dozier (1993) demonstrated that a single band depth within the ice absorption feature could be used to derive snow grain size.". If "band" here is for at the wavelength of 1.03 microns only, consider using "channel"?

The band depth can be defined at any wavelength where an absorption feature is present. Nolin and Dozier (1993) only use a single wavelength, whereas Nolin and Dozier (2000) and the current study use a range of wavelengths between 0.95 microns and 1.09 microns (Eq. 1). To make this clearer, we propose the following changes:

Reflectance in this feature decreases as snow grain size increases (Figure 1), leading to an increase in depth of the absorption feature at the wavelengths between 0.95 µm and 1.09 µm.

Preliminary research by Nolin and Dozier (1993) demonstrated that channel depth at 1.04 µm could be used to derive snow grain size…

Page 3, line 5: Is "absorption feature" and "continuum reflectance" the same concept here?

These terms are related, but they are not the same. "Absorption feature" refers to a localized decrease in surface reflectance over a range of wavelengths, whereas "continuum reflectance" is the surface reflectance expected in the absence of said absorption feature. We make the following change to make this clearer:

This quantity, also known as band depth, is the difference between reflectance without the absorption feature (continuum reflectance) and the observed reflectance with ice absorption.

Page 3, line 8: "Nolin and Dozier (2000) accounted for the latter issue by scaling band depth relative to the continuum reflectance, which is linearly interpolated between 0.95 µm and 1.09 µm." Here, it seems "continuum reflectance" is just a linear line between 0.95 µm and 1.09 µm?

This is correct – a linear line between the bounds of the absorption feature is a reasonable approximation for the continuum reflectance at these wavelengths. We propose to make the following change:

Nolin and Dozier (2000) accounted for the latter issue by scaling band depth relative to the continuum reflectance, for which the authors found that linear interpolation between 0.95 µm and 1.09 µm is a reasonable approximation.

Page 4, line 7: "Band area is computed from an observation of spectral reflectance and best matched to a band area within a lookup table or to a calibration curve of modeled band areas." Here, it would be helpful to explain what does calibration curve looks like? For example, is band area a function of grain size?

This is a good suggestion, and we propose to add the following:

Band area is computed from an observation of spectral reflectance and best matched to a band area within a lookup table or to a calibration curve of modeled band areas. For the purposes of this study, calibration curves are piecewise polynomials that approximate band area as a function of snow grain size.

ND2000 technique was well-documented in the original paper, and readers will likely be able to understand these concepts as they continue reading this paper. But before the authors dive into bias analyses, plots illustrating "continuum reflectance" and "calibration curve" (like Figures 10 and 11) would be helpful here.

Figures 4-7 and 11 provide several visual examples of calibration curves look, so the only addition we propose is the additional text given in the comment above. We agree that an additional plot of band area would be beneficial, so we propose to add a second plot to Figure 1. The updated figure and caption are given below, as is updated text for Page 3, Lines 5-6 and Lines 13-15.

[Figure]

*Figure 1. (a) The spectral dependence of snow directional-hemispherical albedo as a function of effective snow grain size, as derived by SNICAR. The reflectance curved were modeled assuming spherical ice particles and a solar zenith angle of 60°. (b) Close-up view of the ice absorption feature centered at 1.03 µm with $r_{eff}$ = 250 µm. The dashed line in the continuum reflectance at the wavelengths 0.95-1.09 µm, and the grey shading is the band area estimated using Eq. 1.*

(Page 3, Lines 4-6) Reflectance in this feature decreases as snow grain size increases (Fig. 1a), leading to an increase in depth of the absorption feature. This quantity, also known as band depth, is the difference between reflectance without the absorption feature (continuum reflectance) and observed reflectance (Fig. 1b).

(Page 3, Lines 13-15) The integrand of Eq. 1 is the scaled band depth at each wavelength within the absorption feature. Fig. 1b shows the region that is used to calculate band depth through Eq. 1.

Page 5, line 5, "We assumed direct sunlight for all simulations." So all the downwelling flux on snow surface is direct solar flux? What about cloudy sky/diffuse light? Is this due to the limitation of the DM2000 technique? Would the impact of all variables on retrieved grain size be smaller/larger under a cloudy sky?

We assumed direct sunlight to simulate realistic conditions for snow grain size retrievals. Typical retrievals are performed with air-/space-borne instruments, which require low cloud cover to observe surface radiance. A preliminary test with SNICAR revealed that diffuse light would have a minimal impact on grain size retrievals, so we did not pursue any further analysis. Instead, we propose to elaborate upon Page 5, Line 5:

We assumed direct sunlight for all simulations to recreate realistic sky cover conditions for snow grain size retrievals.

Figure 3: Since SNICAR is a two-stream model, it is no surprise the reflectances agree pretty well for the angle of 60 degrees. Out of curiosity, what about the other solar incident angles?

Preliminary tests with SNICAR and the Monte Carlo model confirmed that they agreed at all solar zenith angles. We made the following change on Page 7, Line 7:

Preliminary analysis shows that hemispherical albedo derived from the Monte Carlo model agrees very closely with that of SNICAR at the given snow grain sizes and solar zenith angles (Figure 3, at $\theta_0 = 60°$).